# Heavy-element production in a compact object merger observed by JWST

Andrew J. Levan[1,2 ✉], Benjamin P. Gompertz[3,4], Om Sharan Salafia[5,6], Mattia Bulla[7,8,9], Eric Burns[10], Kenta Hotokezaka[11,12], Luca Izzo[13,14], Gavin P. Lamb[15,16], Daniele B. Malesani[1,17,18], Samantha R. Oates[3,4], Maria Edvige Ravasio[1,5], Alicia Rouco Escorial[19], Benjamin Schneider[20], Nikhil Sarin[21,22], Steve Schulze[22], Nial R. Tanvir[16], Kendall Ackley[2], Gemma Anderson[23], Gabriel B. Brammer[17,18], Lise Christensen[17,18], Vikram S. Dhillon[24,25], Phil A. Evans[16], Michael Fausnaugh[20,26], Wen-fai Fong[27,28], Andrew S. Fruchter[29], Chris Fryer[30,31,32,33], Johan P. U. Fynbo[17,18], Nicola Gaspari[1], Kasper E. Heintz[17,18], Jens Hjorth[13], Jamie A. Kennea[34], Mark R. Kennedy[35,36], Tanmoy Laskar[1,37], Giorgos Leloudas[38], Ilya Mandel[39,40], Antonio Martin-Carrillo[41], Brian D. Metzger[42,43], Matt Nicholl[44], Anya Nugent[27,28], Jesse T. Palmerio[45], Giovanna Pugliese[46], Jillian Rastinejad[27,28], Lauren Rhodes[47], Andrea Rossi[48], Andrea Saccardi[45], Stephen J. Smartt[44,47], Heloise F. Stevance[47,49], Aaron Tohuvavohu[50], Alexander van der Horst[33], Susanna D. Vergani[45], Darach Watson[17,18], Thomas Barclay[51], Kornpob Bhirombhakdi[29], Elmé Breedt[52], Alice A. Breeveld[53], Alexander J. Brown[24], Sergio Campana[5], Ashley A. Chrimes[1], Paolo D'Avanzo[5], Valerio D'Elia[54,55], Massimiliano De Pasquale[56], Martin J. Dyer[24], Duncan K. Galloway[39,40], James A. Garbutt[24], Matthew J. Green[57], Dieter H. Hartmann[58], Páll Jakobsson[59], Paul Kerry[24], Chryssa Kouveliotou[33], Danial Langeroodi[13], Emeric Le Floc'h[60], James K. Leung[40,61,62], Stuart P. Littlefair[24], James Munday[2,63], Paul O'Brien[16], Steven G. Parsons[24], Ingrid Pelisoli[2], David I. Sahman[24], Ruben Salvaterra[64], Boris Sbarufatti[5], Danny Steeghs[2,40], Gianpiero Tagliaferri[5], Christina C. Thöne[65], Antonio de Ugarte Postigo[66] & David Alexander Kann[67]

The mergers of binary compact objects such as neutron stars and black holes are of central interest to several areas of astrophysics, including as the progenitors of gamma-ray bursts (GRBs)[1], sources of high-frequency gravitational waves (GWs)[2] and likely production sites for heavy-element nucleosynthesis by means of rapid neutron capture (the r-process)[3]. Here we present observations of the exceptionally bright GRB 230307A. We show that GRB 230307A belongs to the class of long-duration GRBs associated with compact object mergers[4–6] and contains a kilonova similar to AT2017gfo, associated with the GW merger GW170817 (refs. 7–12). We obtained James Webb Space Telescope (JWST) mid-infrared imaging and spectroscopy 29 and 61 days after the burst. The spectroscopy shows an emission line at 2.15 microns, which we interpret as tellurium (atomic mass $A = 130$) and a very red source, emitting most of its light in the mid-infrared owing to the production of lanthanides. These observations demonstrate that nucleosynthesis in GRBs can create r-process elements across a broad atomic mass range and play a central role in heavy-element nucleosynthesis across the Universe.

GRB 230307A was detected by the Fermi Gamma-ray Burst Monitor (GBM) and GECAM at 15:44:06 UT on 7 March 2023 (refs. 13,14). Its measured duration of $T_{90} \approx 35$ s and exceptionally high prompt fluence of $(2.951 \pm 0.004) \times 10^{-3}$ erg cm$^{-2}$ in the 10–1,000-keV band make it the second brightest GRB ever detected and ostensibly a 'long-soft' GRB (Fig. 1).

The burst was also detected by several other high-energy instruments (Methods), enabling source triangulation by the InterPlanetary Network (IPN). The Neil Gehrels Swift Observatory (Swift) tiled the IPN localization[15], which revealed one candidate X-ray afterglow[16]. We obtained optical observations of the field with the ULTRACAM instrument, mounted on the 3.5-m New Technology Telescope (NTT). These observations revealed a new source coincident with the Swift X-ray source and we identified it as the optical afterglow of GRB 230307A (ref. 17). Given the very bright prompt emission, the afterglow is unusually weak (Fig. 1).

We obtained extensive follow-up observations in the optical and near-infrared with the Gemini South telescope and the Very Large Telescope (VLT); in the X-ray with the Swift X-ray Telescope (XRT) and the Chandra X-ray Observatory; and in the radio with the Australia Telescope Compact Array (ATCA) and MeerKAT. Multi Unit Spectroscopic Explorer (MUSE) integral field spectrograph observations provided the redshift of a bright spiral galaxy at $z = 0.0646 \pm 0.0001$

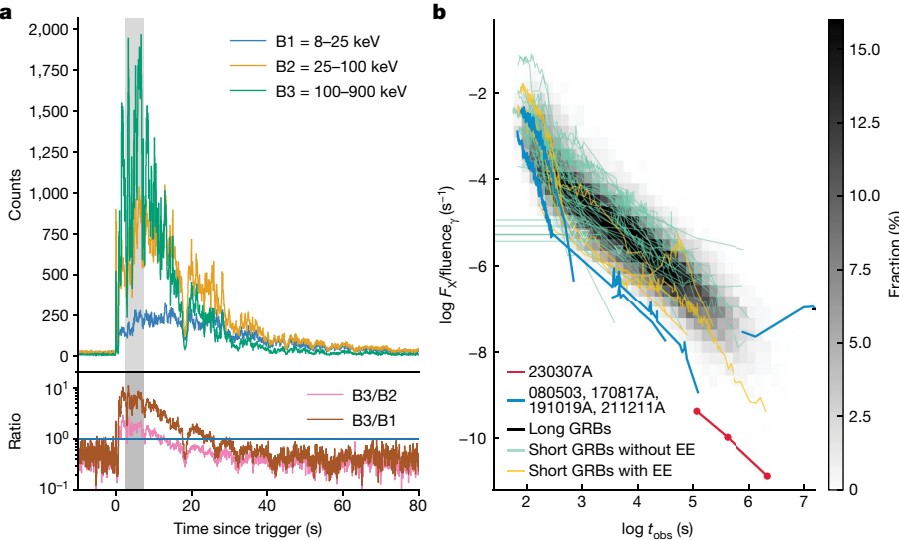

**Fig. 1 | The high-energy properties of GRB 230307A. a**, The light curve of the GRB at 64-ms time resolution with the Fermi/GBM. The shaded region indicates the region in which saturation may be an issue. The burst begins very hard, with the count rate dominated by photons in the hardest (100–900-keV) band, but rapidly softens, with the count rate in the hard band being progressively overtaken by softer bands (such as 8–25 keV and 25–100 keV) beyond about 20 s. This strong hard-to-soft evolution is reminiscent of GRB 211211A (ref. 20) and is caused by the motion of two spectral breaks through the gamma-ray regime (see Methods). **b**, The X-ray light curves of GRBs from the Swift X-ray telescope. These have been divided by the prompt fluence of the burst, which broadly scales with the X-ray light curve luminosity, resulting in a modest spread of afterglows. The greyscale background represents the ensemble of long GRBs. GRB 230307A is an extreme outlier of the >1,000 Swift GRBs, with an extremely faint afterglow for the brightness of its prompt emission. Other merger GRBs from long bursts, and those suggested to be short with extended emission (EE), occupy a similar region of the parameter space. This suggests that the prompt to afterglow fluence could be a valuable tool in distinguishing long GRBs from mergers and those from supernovae.

offset 30.2 arcsec (38.9 kiloparsec in projection) from the burst position (Fig. 2; also ref. 18).

Our ground-based campaign spans 1.4 to 41 days after the burst (Extended Data Tables 1 and 2). At 11 days, infrared observations demonstrated a transition from an early blue spectral slope to a much redder one, consistent with the appearance of a kilonova[3,19]. On the basis of this detection, we requested JWST observations, which were initiated on 5 April 2023. At the first epoch (+28.4 days after the GRB), we took six-colour observations with the Near Infrared Camera (NIRCam) (Fig. 2), as well as a spectrum with the Near Infrared Spectrograph (NIRSpec) covering 0.5–5.5 microns (Fig. 3).

The NIRCam observations reveal an extremely red source with F150W(AB) = 28.11 ± 0.12 mag and F444W(AB) = 24.62 ± 0.01 mag. A faint galaxy is detected in these data, with NIRSpec providing a redshift of $z$ = 3.87, offset approximately 0.3 arcsec from the burst position. The probability of chance alignment for this galaxy and the $z$ = 0.065 spiral are comparable. However, the properties of the burst are inconsistent with an origin at $z$ = 3.87; the implied isotropic equivalent energy release would exceed all known GRBs by an order of magnitude or more, the luminosity and colour evolution of the counterpart would be unlike any observed GRB afterglow or supernova (Supplementary Information). A second epoch of JWST observations was obtained approximately 61 days after the burst. These observations showed that the source had faded by 2.4 mag in F444W, demonstrating a rapid decay expected in a low-redshift kilonova scenario and effectively ruling out alternatives (Supplementary Information). We therefore conclude that GRB 230307A originated from the galaxy at $z$ = 0.065.

Some properties of GRB 230307A are remarkably similar to those of the bright GRB 211211A, which was also accompanied by a kilonova[4–6]. In particular, the prompt emission consists of a hard pulse lasting for approximately 19 s, followed by much softer emission. The prompt emission spectrum is well modelled by a double broken power law with two spectral breaks moving through the gamma-ray band (Methods), suggesting a synchrotron origin of the emission[20]. The X-ray afterglow

is exceptionally faint, much fainter than most bursts when scaled by the prompt GRB fluence (see Fig. 1 and Supplementary Information). The development of the optical and infrared counterpart is also similar to GRB 211211A, with an early blue colour and a subsequent transition to red on a timescale of a few days. In Fig. 4, we plot the evolution of the counterpart compared with the kilonova AT2017gfo (refs. 7–12,21,22), identified in association with the GW-detected binary neutron star merger, GW170817 (ref. 2). AT2017gfo is the most rapidly evolving thermal transient ever observed, much more rapid than supernovae or even fast blue optical transients[23]. The counterpart of GRB 230307A seems to show near-identical decline rates to AT2017gfo both at early times in the optical and infrared as well as later in the mid-infrared (ref. 24). These similarities are confirmed by a joint fit of afterglow and kilonova models to our multiwavelength data (Supplementary Information).

The JWST observations provide a detailed view of kilonova evolution. On timescales of roughly 30 days, it is apparent that the kilonova emits almost all of its light in the mid-infrared, beyond the limits of sensitive ground-based observations. This is consistent with some previous model predictions[25]. Notably, despite its powerful and long-lived prompt emission that strongly contrasts GW170817/GRB 170817A, the GRB 230307A kilonova is remarkably similar to AT2017gfo. This was also the case for GRB 211211A (refs. 4–6,26) and suggests that the kilonova signal is relatively insensitive to the GRB.

Our NIRSpec spectrum shows a broad emission feature with a central wavelength of 2.15 microns, visible in both epochs of JWST spectroscopy (Fig. 3). At longer wavelengths, the spectrum shows a slowly rising continuum up to 4.5 microns, followed by either an extra feature or a change of spectral slope. The colours of the counterpart at this time can be explained by kilonova models (Supplementary Information).

A similar emission-like feature is also visible in the later epochs of X-shooter observations of AT2017gfo (ref. 9), measured at 2.1 microns in ref. 27. Furthermore, the late-time mid-infrared emission and colours are consistent with those observed with AT2017gfo with Spitzer[24]. These similarities further strengthen both the kilonova interpretation

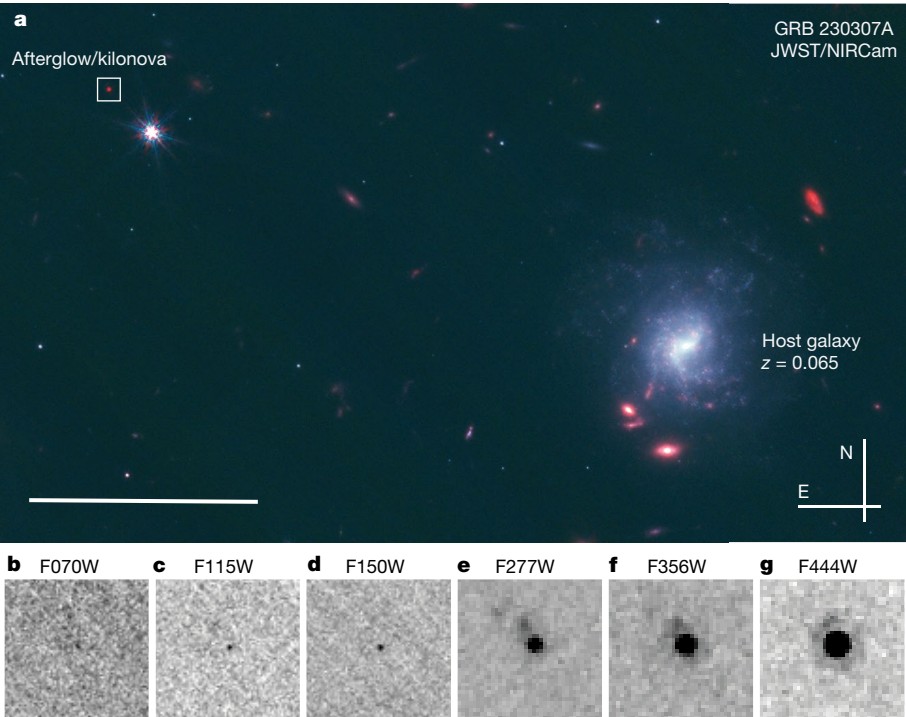

**Fig. 2 | JWST images of GRB 230307A at 28.5 days post burst. a,** The wide-field image combining the F115W, F150W and F444W images. The putative host is the bright face-on spiral galaxy, whereas the afterglow appears at a 30-arcsec offset, within the white box. The scale bar at the lower left represents 10″. **b–g,** Cut-outs of the NIRCam data around the GRB afterglow location. The source is faint and barely detected in the bluer bands but very bright and well detected in the red bands. In the red bands, a faint galaxy is present northeast of the transient position. This galaxy has a redshift of $z = 3.87$ but we consider it to be a background object unrelated to the GRB (see Supplementary Information).

and the redshift measurement of GRB 230307A (Fig. 3). We interpret this feature as arising from the forbidden [Te III] transition between the ground level and the first fine-structure level of tellurium, with an experimentally determined wavelength of 2.1019 microns (ref. 28). The presence of tellurium is plausible, as it lies at the second peak in the $r$-process abundance pattern, which occurs at atomic masses around

$A \approx 130$ (ref. 29). Therefore, it should be abundantly produced in kilonovae, as seen in hydrodynamical simulations of binary neutron star mergers with nucleosynthetic compositions similar to those favoured for AT2017gfo (ref. 30). Furthermore, the typical ionization state of Te in kilonova ejecta is expected to be Te III at this epoch because of the efficient radioactive ionization[31]. Tellurium has recently been suggested

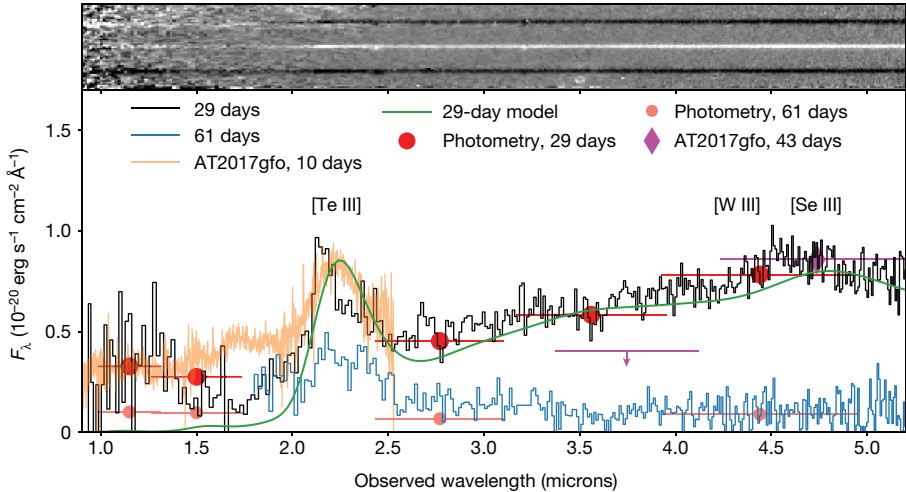

**Fig. 3 | JWST/NIRSpec spectroscopy of the counterpart of GRB 230307A.** The top portion shows the 2D spectrum rectified to a common wavelength scale. The transient is well detected beyond 2 microns but not shortward, indicative of an extremely red source. Emission lines from the nearby galaxy at $z = 3.87$ can also be seen offset from the afterglow trace. The lower panel shows the 1D extraction of the spectrum in comparison with the latest (10-day) AT2017gfo epoch and a kilonova model. A clear emission feature can be seen at about 2.15 microns at both 29 and 61 days. This feature is consistent with the expected location of [Te III], whereas redder features are compatible with lines from [Se III] and [W III]. This line is also clearly visible in the scaled late-time spectrum of AT2017gfo (refs. 27,32), whereas the red colours are also comparable with those of AT2017gfo as measured with Spitzer (ref. 24; shown scaled to the 29-day NIRSpec spectrum). Error bars on photometry refer to the $1\sigma$ error bar on the $y$ axis and the filter width on the $x$ axis.

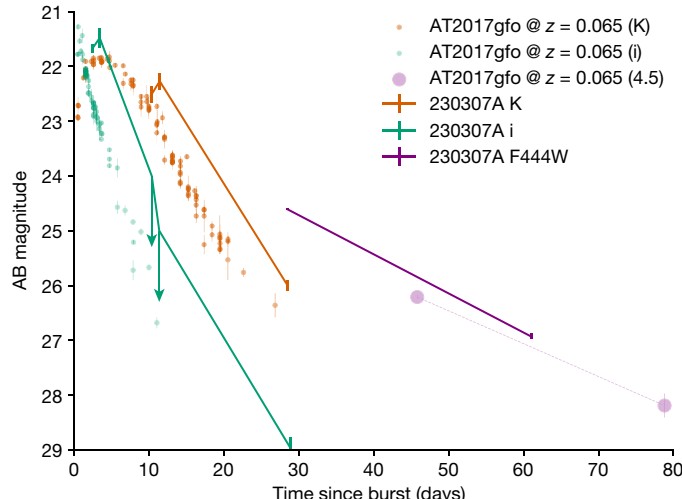

**Fig. 4 | A comparison of the counterpart of GRB 230307A with AT2017gfo associated with GW170817.** AT2017gfo has been scaled to the same distance as GRB 230307A. Beyond about 2 days, the kilonova dominates the counterpart. The decay rates in both the optical and infrared are very similar to those in AT2017gfo. These are too rapid for any plausible afterglow model. There is also good agreement in the late-time slope between the measurements made at 4.4 microns with the JWST and at 4.5 microns for AT2017gfo with Spitzer[24]. Error bars refer to the $1\sigma$ uncertainty.

as the origin of the same feature in the spectrum of AT2017gfo (ref. 32). A previous study[27] also identified this tellurium transition and noted that the observed feature is most likely two blended emission lines. Tellurium can also be produced by means of the slower capture of neutrons in the s-process. Indeed, this line is also seen in planetary nebulae[33]. The detection of [Te III] 2.1 μm provides an extra r-process element, building on the earlier detection of strontium[34]. Notably, although strontium is a light r-process element associated with the so-called first peak, tellurium is a heavier second-peak element, requiring different nucleosynthetic pathways. The mass of Te III estimated from the observed line flux is about $10^{-3}\,M_\odot$ (Supplementary Information). Although weaker, we also note that the spectral feature visible at 4.5 microns is approximately consistent with the expected location of the first-peak element selenium and the near-third-peak element tungsten[35]. In future events, further elemental lines can be used to resolve this difference[35], with very different appearances redward of the NIRSpec cut-off (5.5 microns). For nearby kilonovae, observations should also be plausible by the JWST with the Mid-Infrared Instrument (MIRI) out to 15 microns.

Detailed spectral fitting at late epochs is challenging because of the breakdown of the assumptions about local thermodynamic equilibrium, which are used to predict kilonova spectra at earlier ages, as well as fundamental uncertainties in the atomic physics of r-process elements. However, these observations provide a calibration sample for informing future models. The red continuum emission indicates large opacity in the mid-infrared at low temperatures, for example, about $10\,cm^2\,g^{-1}$ at around 700 K, which may suggest that lanthanides (atomic numbers 58–71) are abundant in the ejecta.

The host galaxy is a low-mass system (about $2.5 \times 10^9\,M_\odot$) dominated by an old population. The large offset is consistent with the largest offsets seen in short GRBs[36,37] and could be attained by a binary with a velocity of a few hundred km s$^{-1}$ and a merger time $>10^8$ years. Alternatively, the faint optical/infrared detection of the source at the second JWST observation could be because of an underlying globular cluster host, which could create compact object systems at enhanced rates through dynamical interactions[38].

It is notable that GRB 230307A is an extremely bright GRB, with only the exceptional GRB 221009A being brighter[39]. The detection of kilonovae in two of the ten most fluent Fermi/GBM GRBs implies that mergers may contribute substantially to the bright GRB population (see Supplementary Information). Indeed, several further long GRBs, including GRB 060614 (refs. 40,41), GRB 111005A (ref. 42) and GRB 191019A (ref. 43), have been suggested to arise from mergers. If a substantial number of long GRBs are associated with compact object mergers, they provide an essential complement to GW detections. First, joint GW–GRB detections, including long GRBs, can push the effective horizons of GW detectors to greater distances and provide much smaller localizations[4,44]. Second, long GRBs can be detected without GW detectors, providing a valuable route for enhancing kilonova detections. Third, JWST can detect kilonova emission at redshifts substantially beyond the horizons of the current generation of GW detectors, enabling the study of kilonovae across a greater volume of the Universe.

The duration of the prompt gamma-ray emission in these mergers remains challenging to explain. In particular, the natural timescales for emission in compact object mergers are much shorter than the measured duration of GRB 230307A. Previously suggested models that may also explain GRB 230307A include magnetars[45], black hole–neutron star mergers[46,47] or even neutron star–white dwarf systems[6]. It has also been suggested that collapsars may power the r-process[48], in which case one may interpret GRB 230307A as an unusual collapsar. However, such a progenitor is not plausible, as there is no star formation at the location of GRB 230307A. The duration problem might become immaterial if the jet timescale does not directly track the accretion timescale in the post-merger system. Such a behaviour has recently been proposed on the basis of insights from general-relativistic magnetohydrodynamical simulations[49,50], which suggest that the duration of the jet can extend up to several times the disk viscous timescale, creating long GRBs from short-lived mergers.

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

¹Department of Astrophysics, Institute for Mathematics, Astrophysics and Particle Physics (IMAPP), Radboud University, Nijmegen, The Netherlands. ²Department of Physics, University of Warwick, Coventry, UK. ³Institute for Gravitational Wave Astronomy, University of Birmingham, Birmingham, UK. ⁴School of Physics and Astronomy, University of Birmingham, Birmingham, UK. ⁵INAF - Osservatorio Astronomico di Brera, Merate, Italy. ⁶INFN - Sezione di Milano Bicocca, Milano, Italy. ⁷Department of Physics and Earth Science, University of Ferrara, Ferrara, Italy. ⁸INFN - Sezione di Ferrara, Ferrara, Italy. ⁹INAF - Osservatorio Astronomico d'Abruzzo, Teramo, Italy. ¹⁰Department of Physics & Astronomy, Louisiana State University, Baton Rouge, LA, USA. ¹¹Research Center for the Early Universe, Graduate School of Science, The University of Tokyo, Bunkyo, Japan. ¹²Kavli IPMU (WPI), UTIAS, The University of Tokyo, Kashiwa, Chiba, Japan. ¹³DARK, Niels Bohr Institute, University of Copenhagen, Copenhagen N, Denmark. ¹⁴INAF - Osservatorio Astronomico di Capodimonte, Naples, Italy. ¹⁵Astrophysics Research Institute, Liverpool John Moores University, Liverpool, UK. ¹⁶School of Physics and Astronomy, University of Leicester, Leicester, UK. ¹⁷Cosmic Dawn Center (DAWN), Copenhagen, Denmark. ¹⁸Niels Bohr Institute, University of Copenhagen, Copenhagen N, Denmark. ¹⁹European Space Agency (ESA), European Space Astronomy Centre (ESAC), Madrid, Spain. ²⁰Kavli Institute for Astrophysics and Space Research, Massachusetts Institute of Technology, Cambridge, MA, USA. ²¹Nordita, Stockholm University and KTH Royal Institute of Technology, Stockholm, Sweden. ²²The Oskar Klein Centre, Department of Physics, Stockholm University, AlbaNova University Center, Stockholm, Sweden. ²³International Centre for Radio Astronomy Research, Curtin University, Perth, Western Australia, Australia. ²⁴Department of Physics and Astronomy, University of Sheffield, Sheffield, UK. ²⁵Instituto de Astrofísica de Canarias, La Laguna, Tenerife, Spain. ²⁶Department of Physics & Astronomy, Texas Tech University, Lubbock, TX, USA. ²⁷Center for Interdisciplinary Exploration and Research in Astrophysics, Northwestern University, Evanston, IL, USA. ²⁸Department of Physics and Astronomy, Northwestern University, Evanston, IL, USA. ²⁹Space Telescope Science Institute, Baltimore, MD, USA. ³⁰Center for Theoretical Astrophysics, Los Alamos National Laboratory, Los Alamos, NM, USA. ³¹Department of Astronomy, The University of Arizona, Tucson, AZ, USA. ³²Department of Physics and Astronomy, The University of New Mexico, Albuquerque, NM, USA. ³³Department of Physics, The George Washington University, Washington, DC, USA. ³⁴Department of Astronomy and Astrophysics, The Pennsylvania State University, University Park, PA, USA. ³⁵School of Physics, University College Cork, Cork, Ireland. ³⁶Jodrell Bank Centre for Astrophysics, Department of Physics and Astronomy, The University of Manchester, Manchester, UK. ³⁷Department of Physics & Astronomy, University of Utah, Salt Lake City, UT, USA. ³⁸DTU Space, National Space Institute, Technical University of Denmark, Lyngby, Denmark. ³⁹School of Physics and Astronomy, Monash University, Clayton, Victoria, Australia. ⁴⁰ARC Centre of Excellence for Gravitational Wave Discovery (OzGrav), Monash University, Clayton, Victoria, Australia. ⁴¹School of Physics and Centre for Space Research, University College Dublin, Dublin, Ireland. ⁴²Columbia Astrophysics Laboratory, Department of Physics, Columbia University, New York, NY, USA. ⁴³Center for Computational Astrophysics, Flatiron Institute, New York, NY, USA. ⁴⁴Astrophysics Research Centre, School of Mathematics and Physics, Queen's University Belfast, Belfast, UK. ⁴⁵GEPI, Observatoire de Paris, Université PSL, CNRS, Meudon, France. ⁴⁶Anton Pannekoek Institute for Astronomy, University of Amsterdam, Amsterdam, The Netherlands. ⁴⁷Department of Physics, University of Oxford, Oxford, UK. ⁴⁸INAF - Osservatorio di Astrofisica e Scienza dello Spazio, Bologna, Italy. ⁴⁹Department of Physics, The University of Auckland, Auckland, New Zealand. ⁵⁰Department of Astronomy & Astrophysics, University of Toronto, Toronto, Ontario, Canada. ⁵¹NASA Goddard Space Flight Center, Greenbelt, MD, USA. ⁵²Institute of Astronomy, University of Cambridge, Cambridge, UK. ⁵³Mullard Space Science Laboratory, University College London, Holmbury St. Mary, UK. ⁵⁴Agenzia Spaziale Italiana (ASI) Space Science Data Center (SSDC), Rome, Italy. ⁵⁵INAF - Osservatorio Astronomico di Roma, Rome, Italy. ⁵⁶Department of Mathematics, Physics, Informatics and Earth Sciences, University of Messina, Polo Papardo, Messina, Italy. ⁵⁷School of Physics and Astronomy, Tel Aviv University, Tel Aviv, Israel. ⁵⁸Department of Physics and Astronomy, Clemson University, Clemson, SC, USA. ⁵⁹Centre for Astrophysics and Cosmology, Science Institute, University of Iceland, Reykjavik, Iceland. ⁶⁰CEA, IRFU, DAp, AIM, Université Paris-Saclay, Université Paris Cité, Sorbonne Paris Cité, CNRS, Gif-sur-Yvette, France. ⁶¹Sydney Institute for Astronomy, School of Physics, The University of Sydney, Sydney, New South Wales, Australia. ⁶²CSIRO Space and Astronomy, Epping, New South Wales, Australia. ⁶³Isaac Newton Group of Telescopes, Santa Cruz de La Palma, Spain. ⁶⁴INAF IASF-Milano, Milano, Italy. ⁶⁵Astronomical Institute of the Czech Academy of Sciences, Ondřejov, Czechia. ⁶⁶Artemis, Observatoire de la Côte d'Azur, Université Côte d'Azur, Nice, France. ⁶⁷Hessian Research Cluster ELEMENTS, Giersch Science Center (GSC), Goethe University Frankfurt, Campus Riedberg, Frankfurt am Main, Germany. ✉e-mail: a.levan@astro.ru.nl

## Methods

### Observations

Below we outline the observational data that were used in this paper. Magnitudes are given in the AB system unless stated otherwise. We use cosmology resulting from the Planck observations[51]. All uncertainties are given at the $1\sigma$ level unless explicitly stated.

**Gamma-ray observations.** GRB 230307A was first detected by Fermi/GBM and GECAM at 15:44:06 UT on 7 March 2023 (refs. 13,14). It had a duration of $T_{90} \approx 35$ s and an exceptionally bright prompt fluence of $(2.951 \pm 0.004) \times 10^{-3}$ erg cm$^{-2}$ (ref. 52). The burst fell outside the coded field of view of the Swift Burst Alert Telescope (BAT) and so did not receive a sub-degree localization despite a strong detection. However, detections by Swift, GECAM[14], STIX on the Solar Orbiter[53], AGILE[54], ASTROSAT[55], GRBalpha[56], VZLUSAT[57], Konus-WIND[58] and ASO-HXI[59] enabled an enhanced position by means of the IPN to increasingly precise localizations of 1.948 deg$^2$ (ref. 60), 30 arcmin$^2$ (ref. 61) and, ultimately, to 8 arcmin$^2$ (ref. 15). This was sufficiently small to enable tiling with Swift and ground-based telescopes.

**Fermi/GBM data analysis.** In Fig. 1, we plot the light curve of GRB 230307A as seen by the Fermi/GBM in several bands, built by selecting time-tagged event data, binned with a time resolution of 64 ms. The highlighted time interval of 3–7 s after trigger is affected by data loss owing to the bandwidth limit for time-tagged event data[62].

For the spectral analysis, we made use of the CSPEC data, which have 1,024-ms time resolution. Data files were obtained from the online archive at https://heasarc.gsfc.nasa.gov/W3Browse/fermi/fermigbrst.html. Following the suggestion reported by the Fermi Collaboration[62], we analysed the data detected by NaI 10 and BGO 1, which had a source viewing angle less than 60°, and excluded the time intervals affected by pulse pile-up issues (from 2.5 s to 7.5 s). The data extraction was performed with the public software GTBURST, whereas data were analysed with XSPEC. The background, whose time intervals have been selected before and after the source, was modelled with a polynomial function whose order is automatically found by GTBURST and manually checked. In the fitting procedure, we used inter-calibration factors among the detectors, scaled to the only NaI analysed and free to vary within 30%. We used the PG-statistic, valid for Poisson data with a Gaussian background. The best-fit parameters and their uncertainties were estimated through a Markov chain Monte Carlo approach. We selected the time intervals before and after the excluded period of 2.5–7.5 s owing to instrumental effects. In particular, we extracted two time intervals from 0 to 2.5 s (1.25 s each) and 14 time intervals from 7.5 s to 40.5 s (bin width of 2 s except the last two with integration of 5 s to increase the signal-to-noise ratio), for a total of 16 time intervals. We fitted the corresponding spectra with the two smoothly broken power law function[63,64], which has been shown to successfully model the synchrotron-like spectral shape of bright long GRBs, including the merger-driven GRB 211211A (ref. 20).

From our spectral analysis, we found that all spectra up to about 20 s are well modelled by the two smoothly broken power law function, namely, they are described by the presence of two spectral breaks inside the GBM band (8 keV–40 MeV). In particular, in the time intervals between 7.5 s and 19.5 s, the low-energy break $E_{\mathrm{break}}$ is coherently decreasing from $304.3^{+5.2}_{-2.6}$ keV to $52.1^{+4.3}_{-5.1}$ keV, and the typical $\nu F_\nu$ peak energy $E_{\mathrm{peak}}$ is also becoming softer, moving from approximately 1 MeV to 450 keV. The spectral indices of the two power laws below and above the low-energy break are distributed around the values of −0.82 and −1.72, which are similar to the predictions for synchrotron emission in marginally fast-cooling regime (that is, −2/3 and −3/2). This is consistent with what has been found in GRB 211211A (ref. 20). We notice, however, that—in all spectra—the high-energy power law above $E_{\mathrm{peak}}$ is characterized by a much softer index (with a mean value of −4.10 ± 0.24) with respect to the value of roughly −2.5 typically found in Fermi GRBs.

This suggests that the spectral data might require a cut-off at high energy, although further investigations are needed to support this. From 19.5 s until 40.5 s (the last time interval analysed), all the break energies are found to be below 20 keV, close to the GBM low-energy threshold. In the same time intervals, the peak energy $E_{\mathrm{peak}}$ decreases from $682.4^{+3.2}_{-6.1}$ keV to $123.1^{+5.4}_{-4.9}$ keV, and the index of the power law below the peak energy is fully consistent (mean value of −1.45 ± 0.06) with the synchrotron predicted value of −1.5.

**Optical observations. NTT: afterglow discovery.** Following the refinement of the IPN error box to an area of 30 arcmin$^2$ (ref. 61), we obtained observations of the field of GRB 230307A with the ULTRACAM instrument[65], mounted on the 3.5-m NTT at La Silla, Chile. The instrument obtains images in three simultaneous bands and is optimized for short-exposure, low-dead-time observations[65]. We obtained ten 20-s exposures in two pointings in each of the Super SDSS $u$, $g$ and $r$ bands (for which the Super SDSS bands match the wavelength range of the traditional SDSS filters but with a higher throughput[66]). The observations began at 01:53:21 UT on 9 March 2023, approximately 34 h after the GRB. The images were reduced through the HIPERCAM pipeline[66] using bias and flat frames taken on the same night. Visual inspection of the images compared with those obtained with the Legacy Survey[67] revealed a new source coincident with an X-ray source identified through Swift/XRT observations[16], and we identified it as the likely optical afterglow of GRB 230307A (ref. 17). The best available optical position of this source (ultimately measured from our JWST observations, see below) is RA(J2000) = 04 h 03 min 26.02 s, dec.(J2000) = −75° 22′ 42.76″, with an uncertainty of 0.05 arcsec in each axis (Supplementary Fig. 1).

This identification was subsequently confirmed through observations from several other observatories, including refs. 18,68–72. We acquired two further epochs of observations with ULTRACAM on the following nights with ten 20-s exposures in the Super SDSS $u$, $g$ and $i$ bands. Aperture photometry of the source is reported in Extended Data Table 1 and is reported relative to the Legacy Survey for the $g$, $r$ and $i$ bands and to SkyMapper for the $u$ band.

**TESS.** The prompt and afterglow emission of GRB 230307A was detected by the Transiting Exoplanet Survey Satellite (TESS), which observed the field continuously from 3 days before the Fermi trigger to 3 days after at a cadence of 200 s (ref. 73). A reference image was subtracted from the observations to obtain GRB-only flux over this period. The measured flux in the broad TESS filter (600–1,000 nm) is corrected for Galactic extinction and converted to the $I_{\mathrm{c}}$ band assuming a power-law spectrum with $F \propto \nu^{-0.8}$. We then bin the light curve logarithmically, taking the mean flux of the observations in each bin and converting to AB magnitudes. A systematic error of 0.1 mag was added in quadrature to the measured statistical errors to account for the uncertainties in the data processing. These data are presented in Extended Data Table 1.

**Swift/UVOT.** The Swift Ultraviolet/Optical Telescope (UVOT[74]) began observing the field of GRB 230307A about 84.6 ks after the Fermi/GBM trigger[13]. The source counts were extracted using a source region of 5 arcsec radius. Background counts were extracted using a circular region of 20 arcsec radius located in a source-free part of the sky. The count rates were obtained from the image lists using the Swift tool UVOTSOURCE. A faint catalogued unrelated source also falls within the 5 arcsec radius; this will affect the photometry, particularly at late times. We therefore requested a deep template image in white to estimate the level of contamination. We extracted the count rate in the template image using the same 5 arcsec radius aperture. This was subtracted from the source count rates to obtain the afterglow count rates. The afterglow count rates were converted to magnitudes using the UVOT photometric zero points[75,76].

**Gemini.** We obtained three epochs of K-band observations using the FLAMINGOS-2 instrument on the Gemini South telescope. These observations were reduced through the DRAGONS pipeline to produce dark

and sky-subtracted and flat-fielded images[77]. At the location of the optical counterpart to GRB 230307A, we identify a relatively bright K-band source in the first and second epochs, with only an upper limit in epoch 3. We report our photometry, performed relative to secondary standards in the VISTA Hemisphere Survey[78], in Extended Data Table 1.

**VLT imaging.** We carried out observations of the GRB 230307A field with the 8.2-m VLT located in Cerro Paranal, Chile. The observations were obtained with the FORS2 camera (mounted on the Unit Telescope 1, UT1, Antu) in $B$, $R$, $I$ and $z$ bands at several epochs and with the HAWK-I instrument (mounted on the Unit Telescope 4, UT4, Yepun) in the $K$ band at one epoch. All images were reduced using the standard European Southern Observatory (ESO) Reflex pipeline[79]. The source was detected in the FORS2 $z$-band image at about 6.4 days after the Fermi/GBM detection. A single $r'$-band observation of the GRB 230307A was also executed with the 2.6-m VLT Survey Telescope (VST) after 2.37 days from the GRB discovery. In later observations, the source was not detected (see Supplementary Information) and the upper-limit values at the $3\sigma$ level are reported in Extended Data Table 1.

**VLT spectroscopy.** To attempt to measure the redshift of GRB 230307A and the nearby candidate host galaxies, we obtained spectroscopy with the VLT using both the X-shooter and MUSE instruments, mounted, respectively, on the Unit Telescope 3 (UT3, Melipal) and on the UT4 (Yepun).

X-shooter spectroscopy, covering the wavelength range 3,000–22,000 Å, was undertaken on 15 March 2023. Observations were taken at a fixed position angle, with the slit centred on a nearby bright star. X-shooter data have been reduced with standard esorex recipes. Given that only two of the four nod exposures were covering the GRB position, resulting in a total exposure time of 2,400 s on-source, we reduced each single exposure using the stare mode data reduction. Then, we stacked the two 2D frames covering the GRB position using dedicated post-processing tools developed in a Python framework[80].

We further obtained observations with the MUSE integral field unit on 23 March 2023. The MUSE observations cover several galaxies in the field, as well as the GRB position, and cover the wavelength range 4,750–9,350 Å. MUSE data were reduced using standard esorex recipes embedded in a single Python script that performs the entire data-reduction procedure. Later, the resulting datacube was corrected for sky emission residuals using ZAP (ref. 81). The MUSE observations reveal the redshifts for a large number of galaxies in the field, including a prominent spiral G1 at $z = 0.0646$ (see also ref. 18) and a group of galaxies, G2, G3 and G4, at $z = 0.263$; details are provided in Extended Data Table 3.

**X-ray afterglow.** Swift began tiled observations of the IPN localization region with its XRT[82] at 12:56:42 on 8 March 2023 (ref. 83) (https://www.swift.ac.uk/xrt_products/TILED_GRB00110/). XRT made the first reported detection of the afterglow (initially identified as 'Source 2') with a count rate of $0.019 \pm 0.004$ cts$^{-1}$ (ref. 16) and later confirmed it to be fading with a temporal power-law index of $1.1^{+0.6}_{-0.5}$ (ref. 84). XRT data were downloaded from the UK Swift Science Data Centre (UKSSDC[85,86]).

We further obtained observations with the Chandra X-ray observatory (programme ID 402458; PI: Fong/Gompertz). A total of 50.26 ks (49.67 ks of effective exposure) of data were obtained in three visits between 31 March 2023 and 2 April 2023. The source was placed at the default aim point on the S3 chip of the ACIS detector. At the location of the optical and X-ray afterglow of GRB 230307A, we detect a total of 12 counts, with an expected background of approximately 1, corresponding to a detection of the afterglow at >5$\sigma$ based on the photon statistics of ref. 87. To obtain fluxes, we performed a joint spectral fit of the Chandra and Swift/XRT data. The best-fitting spectrum, adopting uniform priors on all parameters, is a power law with a photon index of $\Gamma = 2.50^{+0.30}_{-0.29}$ when fitting with a Galactic $N_H = 1.26 \times 10^{21}$ cm$^{-2}$

(ref. 88) and zero intrinsic absorption (neither XRT nor Chandra spectra have sufficient signal to noise to constrain any intrinsic absorption component). The resultant flux in the 0.3–10-keV band is $F_X(1.7$ days$) = 4.91^{+0.89}_{-0.79} \times 10^{-13}$ erg cm$^{-2}$ s$^{-1}$ during the XRT observation and $F_X(24.8$ days$) = 1.19^{+0.87}_{-0.62} \times 10^{-14}$ erg cm$^{-2}$ s$^{-1}$ during the Chandra observation. Owing to the low count number, the Chandra flux posterior support extends to considerably below the reported median, with the 5th percentile being as low as $F_{X,5th} = 3 \times 10^{-15}$ erg cm$^{-2}$ s$^{-1}$. If a uniform-in-the-logarithm prior on the flux were adopted, this would extend to even lower values. Chandra and XRT fluxes are converted to 1 keV flux densities using the best-fit spectrum (Extended Data Table 2).

**ATCA.** Following the identification of the optical afterglow[89], we requested Target of Opportunity (ToO) observations of GRB 230307A (proposal identification CX529) with the ATCA to search for a radio counterpart. These data were processed using MIRIAD[90], which is the native reduction software package for ATCA data using standard techniques. Flux and bandpass calibration were performed using PKS 1934-638, with phase calibration using interleaved observations of 0454-810.

The first observation took place on 12 March 2023 at 4.46 days post-burst, which was conducted using the 4-cm dual receiver with frequencies centred at 5.5 GHz and 9 GHz, each with a 2 GHz bandwidth. The array was in the 750C configuration (https://www.narrabri.atnf.csiro.au/operations/array_configurations/configurations.html) with a maximum baseline of 6 km. A radio source was detected at the position of the optical afterglow at 9 GHz with a flux density of $92 \pm 22$ μJy but went undetected at 5.5 GHz ($3\sigma$ upper limit of 84 μJy). Two further follow-up observations were also obtained, swapping between the 4-cm and 15-mm dual receivers (the latter with central frequencies of 16.7 GHz and 21.2 GHz, each with a 2 GHz bandwidth). During our second epoch at 10.66 days, we detected the radio counterpart again, having become detectable at 5.5 GHz with marginal fading at 9 GHz. By the third epoch, the radio afterglow had faded below detectability. We did not detect the radio transient at 16.7 GHz or 21.2 GHz in either epoch. All ATCA flux densities are listed in Extended Table 2.

**MeerKAT.** We were awarded time to observe the position of GRB 230307A with the MeerKAT radio telescope through a successful Director's Discretionary Time proposal (PI: Rhodes, DDT-20230313-LR-01). The MeerKAT radio telescope is a 64-dish interferometer based in the Karoo Desert, Northern Cape, South Africa[91]. Each dish is 12 m in diameter and the longest baseline is about 8 km, allowing for an angular resolution of roughly 7 arcsec and a field of view of 1 deg$^2$. The observations we were awarded were made at both L and S bands.

GRB 230307A was observed over three separate epochs between seven and 41 days post-burst. The first two observations were made at both L and S4 bands (the highest frequency of the five S-band sub-bands), centred at 1.28 GHz and 3.06 GHz with bandwidths of 0.856 GHz and 0.875 GHz, respectively. Each observation spent two hours at L band and 20 min at S4 band. The final observation was made only at S4 band with 1 h on target. Please see the paper by Max Planck Institute for Radio Astronomy (MPIfR) for further details on the new MeerKAT S-band receiver.

Each observation was processed using OXKAT, a series of semiautomated Python scripts designed specifically to deal with MeerKAT imaging data[92]. The scripts average the data and perform flagging on the calibrators, from which delay, bandpass and gain corrections are calculated and then applied to the target. The sources J0408-6545 and J0252-7104 were used at the flux and complex gain calibrators, respectively. Flagging and imaging of the target field are performed. We also perform a single round of phase-only self-calibration. We do not detect a radio counterpart in any epoch in either band. The root mean square noise in the field was measured using an empty region of the sky and used to calculate $3\sigma$ upper limits, which are given in Extended Data Table 2.

**JWST observations.** We obtained two epochs of observations of the location of GRB 230307A with the JWST. The first on 5 April 2023, with observations beginning at 00:16 UT (MJD = 60039.01), 28.4 days after the burst (under programme GO 4434; PI: Levan), and the second on 8 May 2023, 61.5 days after the burst (programme 4445; PI: Levan). The observations were at a post-peak epoch because the source was not in the JWST field of regard at the time of the burst and only entered it on 2 April 2023.

At the first epoch, we obtained observations in the F070W, F115W, F150W, F277W, F356W and F444W filters of NIRCam[93], as well as a prism spectrum with NIRSpec[94]. In the second epoch, we obtained NIRCam observations in F115W, F150W, F277W and F444W and a further NIRSpec prism observation. However, in the second epoch, the prism observation is contaminated by light from the diffraction spike of a nearby star and is of limited use, in particular at the blue end of the spectrum. We therefore use only light redward of 1.8 microns. However, even here, we should be cautious in interpreting the overall spectral shape. The feature at 2.15 microns is visible in both the 29-day and 61-day spectra.

We reprocessed and redrizzled the NIRCam data products to remove $1/f$ striping and aid point-spread-function recovery, with the final images having plate scales of 0.02 arcsec per pixel (blue channel) and 0.04 arcsec per pixel (red channel).

In the NIRCam imaging, we detect a source at the location of the optical counterpart of GRB 230307A. This source is weakly detected in all three bluer filters (F070W, F115W and F150W), but is at high signal-to-noise ratio in the redder channels (see Fig. 2). The source is compact and unresolved. We also identify a second source offset (H1) approximately 0.3 arcsec from the burst location. This source is also weakly or non-detected in the bluer bands, and is brightest in the F277W filter.

Because of the proximity of the nearby star and a contribution from diffraction spikes close to the afterglow position, we model point spread functions for the appropriate bands using WebbPSF (ref. 95) and then scale and subtract these from the star position. Photometry is measured in small (0.05 arcsec (blue) and 0.1 arcsec (red)) apertures and then corrected using tabulated encircled energy corrections. As well as the direct photometry of the NIRCam images, we also report a K-band point based on folding the NIRSpec spectrum (see below), through a Two Micron All-Sky Survey (2MASS) Ks filter. This both provides a better broadband spectral energy distribution and a direct comparison with ground-based K-band observations. Details of photometric measurements are shown in Extended Data Table 1

For NIRSpec, we use the available archive-processed level 3 2D spectrum (Fig. 3). In this spectrum, we clearly identify the trace of the optical counterpart, which seems effectively undetected until 2 microns and then rises rapidly. We also identify two likely emission lines that are offset from the burst position. These are consistent with the identification with Hα and [O III] (4959/5007) at a redshift of $z = 3.87$. Both of these lines lie within the F277W filter in NIRCam and support the identification of the nearby source as the origin of these lines.

We extract the spectrum in two small (two-pixel) apertures. One of these is centred on the transient position, whereas the other is centred on the location of the emission lines. Because the offset between these two locations is only about 0.3 arcsec, there is naturally some contamination of each spectrum with light from both sources, but this is minimized by the use of small extraction apertures. The counterpart spectra are shown in Fig. 3. The counterpart is very red, with a sharp break at 2 microns and an apparent emission feature at 2.15 microns. The spectrum then continues to rise to a possible second feature (or a change in the associated spectral slope) at around 4.5 microns.

## Data availability

JWST data are directly available from the MAST archive at archive.stsci. edu. ESO data can be obtained from archive.eso.org and Gemini data

from archive.gemini.edu. Core reduced optical and infrared products can also be downloaded directly from the Electronic Research Data Archive at the University of Copenhagen (ERDA) at https://sid.erda.dk/sharelink/b35FULIcV5. This research has made use of Fermi data, which are publicly available and can be obtained through the High Energy Astrophysics Science Archive Research Center (HEASARC) website at https://heasarc.gsfc.nasa.gov/W3Browse/fermi/fermigbrst.html. Swift data can be obtained from http://www.swift.ac.uk/xrt_curves and Chandra observations from https://cda.harvard.edu/chaser/.

## Code availability

Much analysis for this paper has been undertaken with publicly available codes and the details required to reproduce the analysis are contained in the manuscript.

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

**Acknowledgements** We dedicate this paper to David Alexander Kann, who passed on 10 March 2023. He was the first to realize the exceptional brightness of GRB230307A, and the final messages he sent were about its follow-up. We hope it would satisfy his curiosity to know the final conclusions. This work is based on observations made with the NASA/ESA/CSA James Webb Space Telescope. The data were obtained from the Mikulski Archive for Space Telescopes (MAST) at the Space Telescope Science Institute (STScI), which is operated by the Association of Universities for Research in Astronomy, Inc., under NASA contract NAS 5-03127 for the JWST. These observations are associated with programme nos. 4434 and 4445. This paper is partly based on observations collected at the European Southern Observatory under ESO programme 110.24CF (PI: Tanvir) and on observations obtained at the international Gemini Observatory (programme ID GS-2023A-DD-105), a programme of NOIRLab, which is managed by the Association of Universities for Research in Astronomy (AURA) under a cooperative agreement with the National Science Foundation (NSF) on behalf of the Gemini Observatory partnership: the NSF (USA), National Research Council (Canada), Agencia Nacional de Investigación y Desarrollo (Chile), Ministerio de Ciencia, Tecnología e Innovación (Argentina), Ministério da Ciência, Tecnologia, Inovações e Comunicações (Brazil) and Korea Astronomy and Space Science Institute (Republic of Korea). Processed using the Gemini IRAF package and DRAGONS (Data Reduction for Astronomy from Gemini Observatory North and South). A.J.L., D.B.M. and N.R.T. were supported by the European Research Council (ERC) under the European Union's Horizon 2020 research and innovation programme (grant agreement no. 725246). M.B. acknowledges the Department of Physics and Earth Science of the University of Ferrara for the financial support through the FIRD 2022 grant. K.H. is supported by JST FOREST Program (JPMJFR2136) and the JSPS Grant-in-Aid for Scientific Research (20H05639, 20H00158, 23H01169, 20K14513). G.P.L. is supported by a Royal Society Dorothy Hodgkin Fellowship (grant nos. DHF-R1-221175 and DHF-ERE-221005). M.E.R. acknowledges support from the research programme Athena with project number 184.034.002, which is financed by the Dutch Research Council (NWO). N.S. is supported by a Nordita fellowship. Nordita is supported in part by NordForsk. S.S. acknowledges support from the G.R.E.A.T. research environment, funded by Vetenskapsrådet, the Swedish Research Council, project number 2016-06012. V.S.D. and ULTRACAM are funded by STFC grant ST/V000853/1. G.L. was supported by a research grant (19054) from VILLUM FONDEN. The Cosmic Dawn Center (DAWN) is funded by the Danish National Research Foundation under grant no. 140. J.P.U.F. is supported by the Independent Research Fund Denmark (DFF-4090-00079) and thanks the Carlsberg Foundation for support. D.W. is co-funded by the European Union (ERC, HEAVYMETAL, 101071865). D.K.G. acknowledges support from the Australian Research Council Centre of Excellence for Gravitational Wave Discovery (OzGrav), through project number CE170100004. N.G. acknowledges support from the NWO under project number 680.92.18.02. K.E.H. acknowledges support from the Carlsberg Foundation Reintegration Fellowship Grant CF21-0103. J.H. and D.L. were supported by a VILLUM FONDEN Investigator grant (project number 16599). B.D.M. is supported in part by the NSF (grant AST-2002577). M.N. is supported by the ERC under the European Union's Horizon 2020 research and innovation programme (grant agreement no. 948381) and by UK Space Agency grant no. ST/Y000692/1. S.J.S. acknowledges funding from STFC grants ST/X006506/1 and ST/T000198/1. H.F.S. is supported by the Eric and Wendy Schmidt AI in Science Postdoctoral Fellowship, a Schmidt Futures programme. A.A.B. acknowledges funding from the UK Space Agency. P.O.B. acknowledges funding from STFC grant ST/W000857/1. D.S. acknowledges funding from STFC grants ST/T000406/1, ST/T003103/1 and ST/X001121/1. A.S. acknowledges support from DIM-ACAV+ and CNES. S.C., P.D.A., B.Sb. and G.T. acknowledge funding from the Italian Space Agency, contract ASI/INAF no. I/004/11/4.

**Author contributions** A.J.L. led the project, including the location of the optical afterglow and kilonova and the JWST observations. B.P.G. first identified the source as a likely compact object merger, was co-PI of the Chandra observations and contributed to analysis and writing. O.S.S. contributed to afterglow and kilonova modelling and led the writing of these sections. M.B. was involved in kilonova modelling, E.Bu. contributed to interpretation, placing the burst in context and high-energy properties. K.H. was involved in kilonova spectral modelling and identified the 2.15-μm feature. L.I. contributed to the X-shooter data analysis, reduced the MUSE data and led the host analysis. G.P.L. contributed to afterglow and kilonova modelling. D.B.M. organized the VLT observations and contributed to the data analysis. M.E.R. analysed the Fermi data. A.R.E. analysed the Chandra observations. B.Sc. reduced and analysed VLT observations. N.S. contributed to afterglow and kilonova modelling. S.S. was responsible for placing the burst afterglow in context and demonstrating its faintness. N.R.T. contributed to observations and interpretation. K.A. was involved in the ULTRACAM observations. G.A. led the ATCA observations. G.B.B. reduced the JWST NIRCam data. L.C. processed and analysed the MUSE observations. V.S.D. is the ULTRACAM PI. J.P.U.F. studied the high-z possibilities. W.-f.F. was the PI on the Chandra observations. C.F. contributed to the theoretical interpretation. N.G. was involved in host analysis. J.T.P. contributed to the JWST spectrum visualization. K.E.H., G.P., A.R., S.D.V., S.C., P.D.A., D.H.H., M.D.P., C.C.T., A.d.U.P. and D.A.K. contributed to ESO observations and discussion. D.W. contributed to spectral and progenitor modelling. M.J.D., P.K., S.P.L., J.M., S.G.P., I.P. and D.I.S. contributed to the ULTRACAM observations. A.S. reduced X-shooter observations. G.L. investigated potential similarities with other transients. A.T., P.A.E., B.Sb. and J.A.K. contributed to the Swift observations. M.F. extracted and flux-calibrated the TESS light curve. S.J.S. analysed the JWST spectral lines. H.F.S. performed the BPASS-hoki-ppxf fits to the integrated MUSE flux and contributed the associated figure and text. All authors contributed to manuscript preparation through contributions to concept development, discussion and text.

**Competing interests** The authors declare no competing interests.

**Additional information**
**Correspondence and requests for materials** should be addressed to Andrew J. Levan.

**Extended Data Table 1 | Optical and infrared observations of the optical counterpart of GRB 230307A**

| Time since GRB (days) | Telescope | Band | Exposure time (s) | Magnitude (AB) | Source |
|---|---|---|---|---|---|
| 0.01 | TESS | $I_C$ | 1600 | $18.63 \pm 0.14$ | This work |
| 0.03 | TESS | $I_C$ | 1600 | $18.15 \pm 0.12$ | This work |
| 0.04 | TESS | $I_C$ | 1600 | $17.98 \pm 0.11$ | This work |
| 0.09 | TESS | $I_C$ | 6400 | $18.06 \pm 0.10$ | This work |
| 0.19 | TESS | $I_C$ | 11200 | $18.41 \pm 0.10$ | This work |
| 0.38 | TESS | $I_C$ | 20800 | $19.23 \pm 0.11$ | This work |
| 0.64 | TESS | $I_C$ | 24000 | $19.61 \pm 0.16$ | This work |
| 0.99 | UVOT | $U$ | 2315 | $> 21.1$ | This work |
| 1.16 | UVOT | White | 692 | $22.25^{+0.23}_{-0.19}$ | This work |
| 1.25 | UVOT | White | 2217 | $22.29^{+0.34}_{-0.26}$ | This work |
| 1.43 | ULTRACAM | $u$ | 200 | $> 19.7$ | This work |
| 1.43 | ULTRACAM | $g$ | 200 | $> 20.7$ | This work |
| 1.43 | ULTRACAM | $r$ | 200 | $20.72 \pm 0.15$ | This work |
| 1.61 | UVOT | White | 6468 | $> 22.60$ | This work |
| 2.37 | VST | $r$ | 360 | $21.84 \pm 0.19$ | This work |
| 2.41 | ULTRACAM | $u$ | 200 | $> 21.2$ | This work |
| 2.41 | ULTRACAM | $g$ | 200 | $22.35 \pm 0.26$ | This work |
| 2.41 | ULTRACAM | $i$ | 200 | $21.68 \pm 0.09$ | This work |
| 2.43 | UVOT | White | 3303 | $> 22.0$ | This work |
| 3.39 | ULTRACAM | $u$ | 200 | $> 20.8$ | This work |
| 3.39 | ULTRACAM | $g$ | 200 | $> 22.6$ | This work |
| 3.39 | ULTRACAM | $i$ | 200 | $21.48 \pm 0.18$ | This work |
| 4.89 | UVOT | White | 14921 | $> 23.6$ | This work |
| 6.42 | FORS2 | $z$ | 1440 | $23.24 \pm 0.11$ | This work |
| 6.42 | FORS2 | $B$ | 1380 | $> 26.10$ | This work |
| 10.34 | FLAMINGOS-2 | $K$ | 840 | $22.51 \pm 0.15$ | This work |
| 10.36 | FORS2 | $I$ | 2400 | $> 24.0$ | This work |
| 11.36 | FORS2 | $I$ | 2400 | $> 25.2$ | This work |
| 11.42 | FLAMINGOS-2 | $K$ | 700 | $22.27 \pm 0.15$ | This work |
| 15.45 | FLAMINGOS-2 | $K$ | 950 | $> 22.1$ | This work |
| 17.38 | FORS2 | $R$ | 3000 | $> 25.2$ | This work |
| 19.37 | FORS2 | $R$ | 3000 | $> 25.8$ | This work |
| 19.38 | HAWK-I | $K$ | 2340 | $> 23.4$ | This work |
| 28.89 | JWST | F070W | 1868 | $28.97 \pm 0.20$ | This work |
| 28.83 | JWST | F115W | 1868 | $28.50 \pm 0.07$ | This work |
| 28.86 | JWST | F150W | 1546 | $28.11 \pm 0.12$ | This work |
| 28.83 | JWST | F277W | 1868 | $26.24 \pm 0.01$ | This work |
| 28.86 | JWST | F356W | 1546 | $25.42 \pm 0.01$ | This work |
| 28.89 | JWST | F444W | 1868 | $24.62 \pm 0.01$ | This work |
| 61.48 | JWST | F115W | 1868 | $29.78 \pm 0.31$ | This work |
| 61.51 | JWST | F150W | 1546 | $29.24 \pm 0.17$ | This work |
| 61.51 | JWST | F277W | 1868 | $28.31 \pm 0.12$ | This work |
| 61.48 | JWST | F444W | 1546 | $26.97 \pm 0.04$ | This work |
| 2.35 | GMOS-S | $r$ | 30 | $22.0 \pm 0.3$ | [69] |
| 2.35 | SOAR | $z$ | 310 | $21.8 \pm 0.3$ | [71] |
| 3.35 | SOAR | $z$ | 600 | $21.8 \pm 0.3$ | [71] |

Errors are given at the 1σ level and limits are given at the 3σ level.

**Extended Data Table 2 | X-ray and radio observations of the afterglow of GRB230307A**

| Time since trigger (days) | Telescope | Frequency (Hz) | Flux density ($\mu$Jy) | Source |
|---|---|---|---|---|
| 1.7 | $Swift$/XRT | $2.42 \times 10^{17}$ | $(6.5 \pm 1.1) \times 10^{-2}$ | This work |
| 9.59 | $Swift$/XRT | $2.42 \times 10^{17}$ | $< 7.14 \times 10^{-3}$ | [83] |
| 24.84 | $Chandra$ | $2.42 \times 10^{17}$ | $1.6^{+1.2}_{-0.8} \times 10^{-3}$ | This work |
| 4.46 | ATCA | $5.5 \times 10^{9}$ | $< 84$ | This work |
| 4.46 | ATCA | $9 \times 10^{9}$ | $92 \pm 22$ | This work |
| 10.66 | ATCA | $16.7 \times 10^{9}$ | $< 114$ | This work |
| 10.66 | ATCA | $21.2 \times 10^{9}$ | $< 165$ | This work |
| 10.69 | ATCA | $5.5 \times 10^{9}$ | $92 \pm 36$ | This work |
| 10.69 | ATCA | $9 \times 10^{9}$ | $83 \pm 26$ | This work |
| 25.55 | ATCA | $16.7 \times 10^{9}$ | $< 81$ | This work |
| 25.55 | ATCA | $21.2 \times 10^{9}$ | $< 219$ | This work |
| 25.59 | ATCA | $5.5 \times 10^{9}$ | $< 63$ | This work |
| 25.59 | ATCA | $9 \times 10^{9}$ | $< 63$ | This work |
| 6.64 | MeerKAT | $1.3 \times 10^{9}$ | $< 390$ | This work |
| 6.75 | MeerKAT | $3.1 \times 10^{9}$ | $< 140$ | This work |
| 15.74 | MeerKAT | $1.3 \times 10^{9}$ | $< 350$ | This work |
| 16.04 | MeerKAT | $3.1 \times 10^{9}$ | $< 120$ | This work |
| 40.89 | MeerKAT | $3.1 \times 10^{9}$ | $< 93$ | This work |

Errors are given at the 1$\sigma$ level and upper limits are given at the 3$\sigma$ level.

**Extended Data Table 3 | Properties of possible host galaxies for GRB 230307A**

| Host candidate | RA | Dec | $r$ | $z$ | offset $('')$ | $P_{chance}$ |
|---|---|---|---|---|---|---|
| H1 | 04:03:26.06 | -75:22:42.5 | $> 29$ | 3.87 | 0.30 | 0.05* |
| G1 | 04:03:18.79 | -75:22:55.0 | 17.6 | 0.0645 | 29.9 | 0.09 |
| G2 | 04:03:27.32 | -75:23:09.3 | 18.6 | 0.2633 | 27.0 | 0.15 |
| G3 | 04:03:25.64 | -75:23:17.0 | 18.8 | 0.2626 | 34.2 | 0.27 |
| G4 | 04:03:16.67 | -75:22:23.2 | 19.4 | 0.2627 | 29.9 | 0.32 |

*Formally, because the galaxy is undetected in the $r$-band, $P_{chance}$ is unbounded. This probability is based on the magnitudes measured at other wavelengths.