## [Peer Review File · Nature]

Manuscript Title: Heavy element production in a compact object merger observed by JWST

Reviewer Comments & Author Rebuttals

Reviewer Reports on the Initial Version:

Referees' comments:

Referee #1 (Remarks to the Author):

This paper describes the discovery of an extremely red counterpart to an exceptionally bright gamma ray burst that is plausibly consistent with the production of r-process elements. The JWST spectrum presented is exquisite and unprecedented. The paper also discusses the multi-wavelength properties highlighting why this GRB is rare. The dedication of this paper to Kann is appropriate and will be appreciated. I recommend publication in Nature. I urge the authors to make the following minor improvements:

1. Page 6 Para 2 - the claim of $z=0.0646$ needs to be further solidified by why the high redshift galaxy that is much closer to this position is unrelated. There is adequate discussion in the supplementary information but a summary sentence or two needs to be moved forward to the main text.

2. Page 7 Para 2 - the conclusion that this is a long duration GRB from a compact object merger is pre-mature. Thus far, the paper has only presented on compelling evidence of r-process element production and this should not be confused with necessarily coming from a compact object merger. Later, on page 9, the authors discuss the multi-wavelength properties and various possible origins including NS+NS, NS+WD, magnetars etc. They should hold off on the source discussion until then.

3. Page 8 Para 1 - the direct comparison to Spitzer data of GW170817 is missing (specifically the data in reference [31] which is the basis of theory papers on this topic). The Spitzer data should also be added to Figure 3 as the longer wavelength and later time make them more directly comparable to the data shown here.

4. Page 8 Para 1 - the authors claim the detection of [Te III] extends the claimed detection of strontium in the photospheric phase of GW170817 - it is unclear how these two elements are related. Please add additional justification.

5. Page 8 Para 1 - the selenium vs. tungsten puzzle of the theory reference [45] is mentioned but it is unclear whether the JWST data helps resolve this puzzle. The theory reference presents additional features that be used to resolve this degeneracy and this point should be further developed here.

6. Page 9 Para 1 - the most striking property of the suppressed X-rays as shown in Figure 7 of the supplementary information should be added to the discussion here

7. Page 9 Para 2 - also discuss collapsars with r-process as in theory paper Siegel et al. and why the authors consider fallback in this scenario is an unlikely explanation for the suppressed emission.

8. Page 9 Para 2 - see arXiv:2309.00038 for another theoretical idea for why compact mergers could produce longer duration GRBs

9. There are a very large number of line transitions of heavy elements. Since the [Te III] identification is so central to this paper, I recommend adding some discussion of a few other possible line transitions that may explain the 2.1 μ m feature and perhaps some other line transitions of [Te III] that could be searched for in future events with higher S/N in the redder bands.

10. Page 9 Para 3 - the reference to the importance of iodine is out-of-place. Why does the claim that [Te III] is seen imply iodine was produced? Similarly the reference to gold, thorium and iodine in the abstract also appears unnecessarily media-motivated. I suggest deleting it from the paper and including these in the press release.

Referee #2 (Remarks to the Author):

This paper presents followup observations of a gamma-ray burst and identify signs of a emission excess. Though the distance to the event is not definitely known, the authors present strong arguments for an association with the nearest galaxy. They conclude that the excess emission is most likely due to a radioactive kilonova similar to the one associated with GW170817. Using spectral observations, they identify an emission peak near 2 microns, similar to one observed in GW170817, which further supports this interpretation.

The paper is well-presented and includes a rather comprehensive analysis of the data, using theoretical techniques that, while admittedly uncertain, represent the leading edge of what can be done. I find the case made for identifying the emission excess with a kilonova to be compelling. This would be only the second (after GW170817) kilonova spectroscopically observed and the first in the mid-infrared at late times. This impact of the results are at a level meriting publication in Nature. This impact of the results are at a level meriting publication in Nature. I make comments below that raise some questions regarding the interpretation which would be helpful to address, however I doubt these concerns will overturn the basic scenario presented.

The authors point out that the low luminosity and rapid evolution of the transient rules out typical GRB SNe like SN1998bw. But other SN have been observed that are significantly dimmer and have a steeply declining light curve (e.g., SN2010X, SN2005E), suggesting a very low ejecta mass of ~ 0.1 Msun. As there is not a robust theoretical expectation of how GRB SN are formed, it is difficult to rule out that there may be diversity in their properties — and it may not be surprising that bright counterparts like 1998bw would be the easiest to detect, while dimmer/faster counterparts might have so far have gone unnoticed.

Theoretically, weak SN have been proposed to be associated, e.g., with ultrastripped massive stars (e.g., Taurus et al 2015) which may eject masses of ~ 0.1 Msun. If the SN also ejects a small ^{56}Ni mass, it will be under-luminous and cool, and the effect of recombination in cool material can reduce the opacities, giving an even more rapid evolution than would be inferred from a model using constant opacity. Given that long GRBs have typically been associated with massive stars, to what extent can we empirically rule out the presence some unusual SN from the data of

GW270307 — e.g., how small a mass and ^{56}Ni mass may be required, and are there any theoretical scenarios where this seems remotely plausible?

One compelling argument for a kilonova over a SN may be the infrared continuum emission, which suggests a high opacity and hence lanthanide rich ejecta. However, collapsars have also been suggested to produce r-process ejecta in a similar way to mergers, through neutron-rich disk winds. I suspect that theoretically, a collapsar interpretation may be unnatural, as their winds are expected to be significantly more massive than mergers and it may require fine tuning for a collapsar to power a luminous GRB while also not ejecting too much mass. However, the modeling of collapsar winds is still limited, and one could imagine that only a small fraction of the wind mass is neutron-rich enough to produce radioactive r-process.

While a collapsar model may have theoretical challenges in explaining the kilonova properties, one could turn the argument around and emphasize that the NS merger model has theoretically challenges in explaining the properties of the GRB. While I agree that the kilonova interpretation, on the whole, seems more plausible, some further discussion of a massive star SN/collapsar origin would be welcome.

The authors may want to address an apparent inconsistency in their light curve/spectral modeling. The continuum emission at day 29 is fit assuming an optically thick photosphere with a blackbody temperature of ~ 670 K. The ~ 2.1 micron feature, on the other hand, is explained as nebular emission from optically thin TeIII, with an estimated electron temperature of ~ 3000 K. I suppose the picture in mind is that ejecta has an inverted temperature structure, with the inner layers cool and optically thick, and the outer layers optically thin and hot. If this is the case, one might expect a temporal evolution where as the ejecta expand, the 2.1 feature becomes more prominent relative to the continuum. Is there any evidence for this in the 29 and 61 day spectra?

The presumption of the analysis is that the mid-IR continuum is due to photospheric emission. But it appears that most detailed opacity calculations in the literature (which include large number of lines from lanthanides) likely fail to produce the high opacities in the mid-IR that are inferred in the semi-analytic models (those models use constant opacity, whereas the true opacity decreases to longer wavelengths). Can we be sure that the emission is from a photosphere and not instead from optically thin nebular lines that blend together to form something quasi-thermal? Does this have impact on the inference of composition? (It may be that lanthanides are required in any case to have enough lines that emit in the mid-IR).

Another factor that could be relevant for the interpretation of the IR transient is dust reprocessing. The authors mention extinction briefly in 4.2 (and again when discussing a high-z interpretation) but it did not seem that their discussion covered the relevant possibilities. Dust is known to form in SN ejecta when the temperature drops to the low values seen here, and molecule formation can also affect the opacities. Are there reasons to discard that some dust/molecule formation occurred in the ejected material and plays an important role in forming the mid-IR continuum? Could this negate the need for heavy-element lanthanides? Could it negate the need for r-process at all (going back to a potential SN interpretation)? Of course, dust formation is not well understood theoretically and even more so in r-process material, but some general considerations of whether low mass dusty ejecta could potentially explain the data would be helpful.

Despite these comments, I'll point out that the authors are sufficiently forthright with the limitations in the current ability to spectroscopically model these events, and one should not expect a definitive spectral interpretation; rather the interesting observations pose challenges for future modeling work.

Author Rebuttals to Initial Comments:

We thank both referee's for their thorough and very constructive reports. These have raised a number of good questions which we have commented on below, and implemented at various points within the text. Within this document responses from us (such as this one) are italicized and underlined. Other text is the referees comments to which we respond.

Referee #1 (Remarks to the Author):

This paper describes the discovery of an extremely red counterpart to an exceptionally bright gamma-ray burst that is plausibly consistent with the production of r-process elements. The JWST spectrum presented is exquisite and unprecedented. The paper also discuss the multi-wavelength properties highlighting why this GRB is rare. The dedication of this paper to Kann is appropriate and will be appreciated. I recommend publication in Nature. I urge the authors to make the following minor improvements:

We thank the referee for their careful reading of the manuscript and the suggested modifications which are very valuable. We have taken these into account in our revised version and our specific responses to the comments raised are provided below.

1. Page 6 Para 2 - the claim of $z=0.0646$ needs to be further solidified by why the high redshift galaxy that is much closer to this position is unrelated. There is adequate discussion in the supplementary information but a summary sentence or two needs to be moved forward to the main text.

We have added some discussion about why we disfavour the $z=3.87$. We agree that arguing for this host association earlier (where the redshift is reported) is premature and have therefore simply reported the redshift where the spectroscopy is introduced and moved the discussion of the redshift later.

2. Page 7 Para 2 - the conclusion that this is a long duration GRB from a compact object merger is pre-mature. Thus far, the paper has only presented on compelling evidence of r-process element production and this should not be confused with necessarily coming from a compact object merger. Later, on page 9, the authors discuss the multi-wavelength properties and various possible origins including NS+NS, NS+WD, magnetars etc. They should hold off on the source discussion until then.

We have removed the final sentences of this paragraph and put them into the discussion as suggested.

3. Page 8 Para 1 - the direct comparison to Spitzer data of GW170817 is missing (specifically the data in reference [31] which is the basis of theory papers on this topic). The Spitzer data should also be added to Figure 3 as the longer wavelength and later time make them more directly comparable to the data shown here.

We agree. We have (briefly given space constraints) included the similarity to the Spitzer observations of AT2017gfo within this paragraph and have added them to Figure 3.

4. Page 8 Para 1 - the authors claim the detection of [Te III] extends the claimed detection of strontium in the photospheric phase of GW170817 - it is unclear how these two elements are related. Please add additional justification.

By “extend” in this case we intended to indicate that this is a further r-process element which has been identified, and that it arises from a separate r-process peak. We have tried to be clearer regarding this within the text because indeed there is no direct relation between the two elements.

5. Page 8 Para 1 - the selenium vs. tungsten puzzle of the theory reference [45] is mentioned but it is unclear whether the JWST data helps resolve this puzzle. The theory reference presents additional features that be used to resolve this degeneracy and this point should be further developed here.

We have noted (as in reference [45]) that further observations with a broader wavelength range are important in solving this scenario. In particular, observations redward with MIRI should be possible for kilonovae as bright as GRB 230307A. These observations should cover additional lines and provide a strong constraints.

6. Page 9 Para 1 - the most striking property of the suppressed X-rays as shown in Figure 7 of the supplementary information should be added to the discussion here

Thank you, we do now highlight this point in paragraph 2 where we introduce the afterglow discovery and note that it is faint. We have some discussion of this in the SI. Unfortunately space constraints limit what can be said in the main article.

7. Page 9 Para 2 - also discuss collapsars with r-process as in theory paper Siegel et al. and why the authors consider fallback in this scenario is an unlikely explanation for the suppressed Emission.

We concur that this is a relevant consideration. We have include a brief discussion in this paragraph and also a separate section in the Supplementary Information where we consider unusual supernovae, including the possibility of r-process production in collapsars. In brief, the location of the burst is not where one would expect to find a massive star (far too large an offset), and the properties of r-process transients with collapsars should not, in general, appear like kilonovae – in our SI discussion we have added a citation to the paper of Barnes & Metzger, who look at this possibility explicitly for GRB 211211A.

8. Page 9 Para 2 - see arXiv:2309.00038 for another theoretical idea for why compact mergers could produce longer duration GRBs

Thank you, we have added this reference, which we concur is very relevant as it may alleviate any concerns regarding bursts of this duration from compact object mergers.

9. There are a very large number of line transitions of heavy elements. Since the [Te III] identification is so central to this paper, I recommend adding some discussion of a few other possible line transitions that may explain the 2.1um feature and perhaps some other line transitions of [Te III] that could be searched for in future events with higher S/N in the redder bands.

We thank the referee for pointing this out. We'd refer to Table 6 in Gillanders et al 2023 for the summary of the candidate lines for the 2.1 um feature. For forbidden lines, among the candidate lines Te and Ba are expected to be the most abundant r-process elements. The [Te III] 2.1um and [Ba II] 2.05um lines arise from the energy levels collisionally accessible at the nebular temperature. However, [Ba II] 2.05 um is an E2 line of which the transition rate is lower by a factor of ~100 than [Te III] 2.1um, and therefore, the Ba II line strength should be lower. In our modeling, the second strongest line around 2.1um is [Ir II] 2.09 um but the strength of this line is about less than 10 percent of [Te III].

The second strongest line of [Te III] is at 2.93 micron. The relative strength of this line to the 2.1 micron line depends on the temperature. The lack of the 2.93 line in the NIRSspec data implies that the electron temperature is low. We have added some further discussion of this. Because of space constraints this is largely in the Supplementary Information.

10. Page 9 Para 3 - the reference to the importance of iodine is out-of-place. Why does the claim that [Te III] is seen imply iodine was produced? Similarly the reference to gold, thorium and iodine in the abstract also appears unnecessarily media-motivated. I suggest deleting it from the paper and including these in the press release.

The idea of this paragraph was primarily to reach to the broader Nature readership (i.e. non astronomers). However, we acknowledge that they are unlikely to read the end of the paper and we have removed this paragraph.

Referee #2 (Remarks to the Author):

This paper presents followup observations of a gamma-ray burst and identify signs of a emission excess. Though the distance to the event is not definitely known, the authors present strong arguments for an association with the nearest galaxy. They conclude that the excess emission is most likely due to a radioactive kilonova similar to the one associated

with GW170817. Using spectral observations, they identify an emission peak near 2 microns, similar to one observed in GW170817, which further supports this interpretation. The paper is well-presented and includes a rather comprehensive analysis of the data, using theoretical techniques that, while admittedly uncertain, represent the leading edge of what can be done. I find the case made for identifying the emission excess with a kilonova to be compelling. There would be only the second (after GW170817) kilonova spectroscopically observed and the first in the mid-infrared at late times. This impact of the results are at a level meriting publication in Nature. This impact of the results are at a level meriting publication in Nature. I make comments below that raise some questions regarding the interpretation which would be helpful to address, however I doubt these concerns will overturn the basic scenario presented.

We thank the referee for their thoughts and comments, and also the detailed exposition of them. As a result we have looked into all of the points and have added the relevant discussion. Because of the space requirements within the main article (which already needed to be shortened) much of this material backing up the conclusions is provided within the Supplementary Information.

The authors point out that the low luminosity and rapid evolution of the transient rules out typical GRB SNe like SN1998bw. But other SN have been observed that are significantly dimmer and have a steeply declining light curve (e.g., SN2010X, SN2005E), suggesting a very low ejecta mass of ~ 0.1 Msun. As there is not a robust theoretical expectation of how GRB SN are formed, it is difficult to rule out that there may be diversity in their properties — and it may not be surprising that bright counterparts like 1998bw would be the easiest to detect, while dimmer/faster counterparts might have so far have gone unnoticed. Theoretically, weak SN have been proposed to be associated, e.g., with ultrastripped massive stars (e.g., Taurus et al 2015) which may eject masses of ~ 0.1 Msun. If the SN also ejects a small ^{56}Ni mass, it will be under-luminous and cool, and the effect of recombination in cool material can reduce the opacities, giving an even more rapid evolution than would be inferred from a model using constant opacity. Given that long GRBs have typically been associated with massive stars, to what extent can we empirically rule out the presence some unusual SN from the data of GW270307 — e.g., how small a mass and ^{56}Ni mass may be required, and are there any theoretical scenarios where this seems remotely plausible?

Indeed it is somewhat surprising given the diversity in GRB properties that the population of supernovae uncovered within them appear to be rather homogeneous. We should clearly keep an open mind regarding the possibility of fainter events as the selection effects would tend to prefer the identification of luminous, long-lived SN 1998bw events.

However, for GRB 230307A we do not believe that a collapsar-like origin is plausible. Perhaps, most strongly, the location strongly suggests that the system is old, rather than a young star. For ejection from the $z=0.065$ galaxy to 40 kpc the timescale is far too long for a core collapse event. Even when allowing for longer delays due to binary interactions it is difficult to get delays much longer than ~ 50 Myr, therefore a massive star would appear to be ruled out at this redshift.

One could propose an alternative redshift than $z=0.065$, but until $z=3.87$ (where essentially one cannot explain the emission, as discussed in detail in the SI) there are no host galaxies where substantial kicks and long-lifetimes wouldn't be necessary to reach the observed location. Indeed, in a higher redshift scenario (but less than $z=3.87$) the supernova would no longer be "faint", and so the selection biases wouldn't necessarily apply.

It is, of course, still relevant to consider if the properties of the transient (not just its location) and if these could be explained other processes, perhaps in older populations (as is the case for SN 2005E). These faint fast supernovae reach peak light a few days after the explosion before falling by $\sim 3-4$ magnitudes over the subsequent 20 days. The fall in the case of GRB 230307A is much more rapid, and does not reveal any underlying Nickel exponential decay. The previous events also show clear emission features in their later time spectra (e.g. the Ca triplet in the red optical) that are clearly not present in GRB 230307A.

It is also possible that the supernova could form via some form of fallback disc. These systems would have a fainter peak magnitude, low nickel yield and lower temperatures at later times (e.g. Fryer et al), although are disfavoured by the galactic location of GRB 230307A. The published models, while cool, are also typically at a higher temperature than inferred for GRB 230307A, although the full parameter space has not been studied in detail.

To address this point (and the one below regarding the r-process) in the text, we have added an additional section to the Supplementary Information.

One compelling argument for a kilonova over a SN may be the infrared continuum emission, which suggests a high opacity and hence lanthanide rich ejecta. However, collapsars have also been suggested to produce r-process ejecta in a similar way to mergers, through neutron-rich disk winds. I suspect that theoretically, a collapsar interpretation may be unnatural, as their winds are expected to be significantly more massive than mergers and it may require fine tuning for a collapsar to power a luminous GRB while also not ejecting too much mass. However, the modeling of collapsar winds is still limited, and one could imagine that only a small fraction of the wind mass is neutron-rich enough to produce radioactive r-process. While a collapsar model may have theoretical challenges in explaining the kilonova properties, one could turn the argument around and emphasize that the NS merger model has theoretical challenges in explaining the properties of the GRB. While I agree that the

kilonova interpretation, on the whole, seems more plausible, some further discussion of a massive star SN/collapsar origin would be welcome.

This is certainly a relevant model to compare. We have briefly added some text in the main article, and, as noted above have included a section addressing it within the Supplementary Information. In short, the location of the burst at such a large distance from any plausible low-z host strongly disfavours an origin in a massive star.

From a purely modelling perspective we also now point to the work of Barnes & Metzger who investigated if collapsar-like models could work for GRB 211211A, which has similar evolution at early times (but of course, no JWST observations) and concluded that the parameter space for this was very small.

The authors may want to address an apparent inconsistency in their light curve/spectral modeling. The continuum emission at day 29 is fit assuming an optically thick photosphere with a blackbody temperature of ~670 K. The ~2.1 micron feature, on the other hand, is explained as nebular emission from optically thin TeIII, with an estimated electron temperature of ~3000 K. I suppose the picture in mind is that ejecta has an inverted temperature structure, with the inner layers cool and optically thick, and the outer layers optically thin and hot. If this is the case, one might expect a temporal evolution where as the ejecta expand, the 2.1 feature becomes more prominent relative to the continuum. Is there any evidence for this in the 29 and 61 day spectra?

Thank you for raising this point. Indeed, this is the model which we are proposing. Although the bluer end of the spectrum is contaminated by light from the nearby star, the line is substantially more prominent at these later times. For example the ratio of the peak line flux to the 4.5 micron flux at 29 days is ~1, but at 61 days is ~2.5. We have added some discussion related to this within the Supplementary Information.

The presumption of the analysis is that the mid-IR continuum is due to photospheric emission. But it appears that most detailed opacity calculations in the literature (which include large number of lines from lanthanides) likely fail to produce the high opacities in the mid-IR that are inferred in the semi-analytic models (those models use constant opacity, whereas the true opacity decreases to longer wavelengths). Can we be sure that the emission is from a photosphere and not instead from optically thin nebular lines that blend together to form something quasi-thermal? Does this have impact on the inference of composition? (It may be that lanthanides are required in any case to have enough lines that emit in the mid-IR).

Indeed, it is correct that most calculations in the literature do not find an optically thick solution at this time. The opacity calculations predict somewhat lower opacities, < 1

cm²/g, at a temperature of ~700K under local thermodynamic equilibrium. We can think of the following solutions for the origin of the mid-IR emission: (1) the line blending of collisionally excited lines, (2) the high opacity due to the non-LTE effects, and (3) fluorescence line blending.

We think that solution (1) is highly unlikely because it requires fine tuning of the strengths of ~100 thin lines such that the superposition of the lines produces a smooth quasi-thermal emission (note that this option was briefly mentioned and ruled out in Section 6.2 of the Methods). The optically thick scenario (solution 2), on the other hand, does not require such fine tuning but the existence of ~ 100 thick lines. In fact, Pognan et al 2022, MNRAS 513, 5174 indicate that the non-LTE effect can enhance the opacity. Even if the optical depth in the mid-IR is less than unity, it may be possible to produce a quasi-thermal IR emission by blending fluorescence lines of lanthanides/actinides that absorb photons at < 2 microns and re-emit photons at longer wavelengths. While we cannot quantitatively show the non-LTE effects we consider it is likely that lanthanides/actinides play a role to produce the quasi-thermal emission in the mid-IR (solution 2 or 3).

It is also true that a similar colour is inferred from the Spitzer observations of AT2017gfo (although these do require some extrapolation of the optical/nIR lightcurves for comparison). This does imply that such behaviour is consistent with the scant observations to date.

Another factor that could be relevant for the interpretation of the IR transient is dust reprocessing. The authors mention extinction briefly in 4.2 (and again when discussing a high-z interpretation) but it did not seem that their discussion covered the relevant possibilities. Dust is known to form in SN ejecta when the temperature drops to the low values seen here, and molecule formation can also affect the opacities. Are there reasons to discard that some dust/molecule formation occurred in the ejected material and plays an important role in forming the mid-IR continuum? Could this negate the need for heavy-element lanthanides? Could it negate the need for r-process at all (going back to a potential SN interpretation)? Of course, dust formation is not well understood theoretically and even more so in r-process material, but some general considerations of whether low mass dusty ejecta could potentially explain the data would be helpful.

We thank the referee for raising this point, which we agree is a relevant consideration. As the referee notes there are no detailed calculations for the formation of dust and/or molecules in r-process material, or indeed in the kind of ejecta that would be necessary to explain the rapidity of the transient. Therefore, any considerations of this kind are naturally speculative. In the r-process scenario there are calculations for dust formation, with the conclusion that they do not efficiently form (Takami et al ApJL 789, L6, 2014). Given the location of the transient we do not consider massive star progenitors likely, but we could

consider dust and molecule formation in the white dwarf - neutron star merger scenario. Again, there are no detailed models for this. The late time JWST spectral shape could potentially be explained by a combination of CO emission and dust for the opacity (Gillanders et al. 2023). However, explaining the earlier reddening (at 10 days) in this scenario may be more difficult, and the CO explanation requires a blue-shift to the CO-lines and a structure which is not comparable to those seen in CO emission in supernovae (although the signal to noise is not ideal). In practice, the lack of any specific models for this emission, and the strong similarity of the observed transient with the only very well sampled kilonova leads us to view the kilonova solution as by far the most economical model (and the one which made a testable prediction for our second epoch of JWST observations). However, we have added some text regarding this possibility in the Supplementary Information, and note that future detailed modelling (or GW detection) may shed further light on this issue.

Despite these comments, I'll point out that the authors are sufficiently forthright with the limitations in the current ability to spectroscopic model these events, and one should not expect a definitive spectral interpretation; rather the interesting observations pose challenges for future modeling work.